# Epigenetic instability may alter cell state transitions and anticancer drug resistance

**Anshul Saini**[ID]**, James M. Gallo**[ID]*

Department of Pharmaceutical Sciences, University at Buffalo, Buffalo, New York, United States of America

* jmgallo@buffalo.edu

## Abstract

Drug resistance is a significant obstacle to successful and durable anti-cancer therapy. Targeted therapy is often effective during early phases of treatment; however, eventually cancer cells adapt and transition to drug-resistant cells states rendering the treatment ineffective. It is proposed that cell state can be a determinant of drug efficacy and manipulated to affect the development of anticancer drug resistance. In this work, we developed two stochastic cell state models and an integrated stochastic-deterministic model referenced to brain tumors. The stochastic cell state models included transcriptionally-permissive and -restrictive states based on the underlying hypothesis that epigenetic instability mitigates lock-in of drug-resistant states. When moderate epigenetic instability was implemented the drug-resistant cell populations were reduced, on average, by 60%, whereas a high level of epigenetic disruption reduced them by about 90%. The stochastic-deterministic model utilized the stochastic cell state model to drive the dynamics of the DNA repair enzyme, methylguanine-methyltransferase (MGMT), that repairs temozolomide (TMZ)-induced O6-methylguanine (O6mG) adducts. In the presence of epigenetic instability, the production of MGMT decreased that coincided with an increase of O6mG adducts following a multiple-dose regimen of TMZ. Generation of epigenetic instability via epigenetic modifier therapy could be a viable strategy to mitigate anticancer drug resistance.

## Author summary

Drug therapy for cancer is an important means to abate the disease and improve patient survival; however, tumors often adapt to drug therapy rendering it ineffective. Most strategies to overcome anticancer drug resistance involve identification of drug resistance-inducing targets that may be met with a new drug, forming a drug combination approach. Ultimately, such drug combinations fail as cancer cells continually adapt and find alternative ways to proliferate. Our strategy offers a new approach that focuses on causing epigenetic instability. Histone proteins and their chemical modifications, which are highly plastic are key determinants of gene transcription. We developed cell state transition models that capture epigenetic plasticity and when unstable drug-resistant cell populations are significantly reduced. The cell state transition models were extended to a specific mechanism of TMZ-induced resistance involving the DNA repair enzyme, MGMT. Once again,

**Data Availability Statement:** All relevant data are within the manuscript and its Supporting Information files.

**Funding:** Funding from the State University of New York Empire Innovation program is acknowledged for support of AS PhD. The funder had no role in

study design, data collection and analysis, decision
to publish, or preparation of the manuscript.

**Competing interests:** The authors have declared
that no competing interests exist.

epigenetic instability reduced MGMT production increasing the formation of O6mG
DNA adducts. The combined models demonstrate the potential of epigenetic instability as
a new strategy to mitigate the development of anticancer drug resistance.

## Introduction

The hallmarks of cancer have evolved from their original inception and indicate the complexity of the disease and the remarkable adaptability of cancer cells to sustain growth under adverse conditions including drug therapy [1,2]. Epigenetic reprogramming or plasticity is part of the adaptability armamentarium of cancer cells that alters gene expression by regulating gene transcription through histone post-translational modifications (PTMs). Thus, disrupted epigenetic mechanisms play a pivotal role in cancer biology that are characterized by epigenetic plasticity and altered cell states [3,4]. There have been a number of investigations that have modeled epigenetic plasticity within the context of specific cell state transitions [5,6]. Miyamoto et al [7] considered how gene regulatory networks and epigenetic feedback influenced differentiated and pluripotent cell states. Epithelial-mesenchymal cell state transitions relevant to cancer have been analyzed by a number of investigators that in one case demonstrated that epigenetic feedback altered the dynamics [8], and in another case, accounted for population heterogeneity that displayed multi-stability and hysteresis [9]. Recently, Gunnarsson et al [10] developed an evolutionary dynamic model that explored epigenetic mechanisms conferring anticancer drug resistance and how epigenetic modifying drugs could alter transitions between drug resistant and drug sensitive states.

In recent years, epigenetic modifiers have been used in attempts to reverse resistance to other drugs through epigenetic modifications [11–16]. The reprogramming capability of the epigenome indicates that cells are susceptible to disequilibrium and it is precisely this instability that may be tapped to offer a therapeutic strategy to improve drug therapy.

We previously presented a deterministic cell state model of mutant isocitrate dehydrogenase-1 (IDH1) gliomas that consisted of quiescent, stem and differentiated glioma cells that could transition between each other reflecting intratumoral heterogeneity (ITH) and cellular adaptation [17]. The model incorporated cell proliferation, death, and state transitions in the context of drug resistance. Model simulations indicated that modulation of cell transition rates based upon fluctuating D-2-hydroxy-glutatarate (D2HG) concentrations, a known epigenetic modifier of this type of brain tumor, partially mitigated the evolution of drug-resistant tumors. However, the intrinsic randomness of cell biology is more appropriately captured by stochastic modeling approaches. Various studies have shown that models based on Markovian stochastic transitions between tumor cell states are consistent with observed attributes of tumor heterogeneity [18–20]. In this investigation, the stochastic cell state models consider epigenetic-mediated transcriptionally-permissive and transcriptionally-restrictive states as precursors to drug resistant states.

Mechanisms of drug resistance are diverse and may be associated with specific drugs as is the case for temozolomide (TMZ), an alkylating agent that is used as the standard of care for patients with glioblastoma multiforme (GBM) [21]. TMZ causes the formation of N3-methyladenine, N7-methylguanine and O6-methylguanine (O6mG) DNA adducts with O6mG considered the most lethal and subject to repair by O6-methylguanine methyltransferase (MGMT) [22]. The stochastic cell state model was integrated with an ordinary differential equation model that characterized the gene transcription and translation of MGMT in the presence of a multiple-dose schedule of TMZ that yielded O6mG adducts. The combined

model showed epigenetic instability limited MGMT production whilst leading to higher O6mG adducts. The mathematical models show that implementation of epigenetic instability may be a strategy to subdue the evolution of drug resistance.

## Results

### Basic single-step model

A solid tumor, and herein referred to as a brain tumor or glioblastoma multiforme (GBM), was considered to consist of multiple cell types or states (Fig 1). The cell populations change over time due to proliferation, death, and state transitions. State transitions may be initiated due to the microenvironment (blood flow and hypoxia) and drug therapy as a means to survive and grow. In the model (Fig 1), at time zero, there are five cell states wherein both differentiated glioma (G) and glioma stem cells (GS) exist in two transcriptional states, either permissive (Tp) or restrictive (Tr) that leads to four cell states (G-Tp, G-Tr, GS-Tp, and GS-Tr). Quiescent (Q) cells comprise a fifth cell state. Drug therapy can also lead to two more cell states, drug resistant glioma (G-R) and glioma stem cells (GS-R). Cells in a particular state can undergo either birth, death or transition to another cell state, except Q cells that may only die or transition. This construction of the model permits an understanding of how epigenetic changes

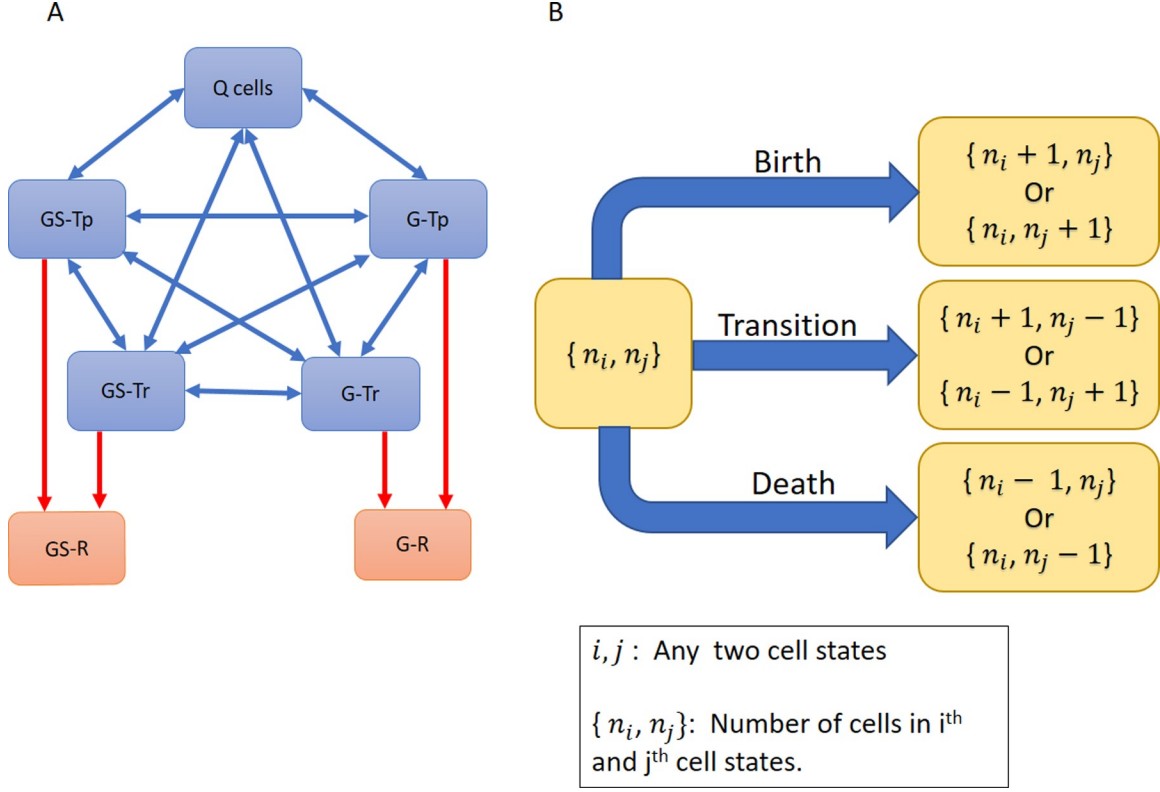

**Fig 1. Schematic of the basic single-step cell state model.** (A) The model consists of five states: Quiescent cells (Q), transcriptionally-permissive Glioma and Glioma Stem cells (G-Tp and GS-Tp, respectively), and transcriptionally-restrictive glioma and glioma stem cells (G-Tr and GS-Tr, respectively). Cells can transition between any two of these states (indicated by two-sided blue arrows). In addition, in the presence of drug therapy, cells may transition to drug resistant states (G-R/GS-R). Once in a resistant cell state, the cells are unable to transition back to drug-sensitive states (indicated by one-way red arrows). (B) Cells can undergo birth (except Q cells), death or transition to another state.

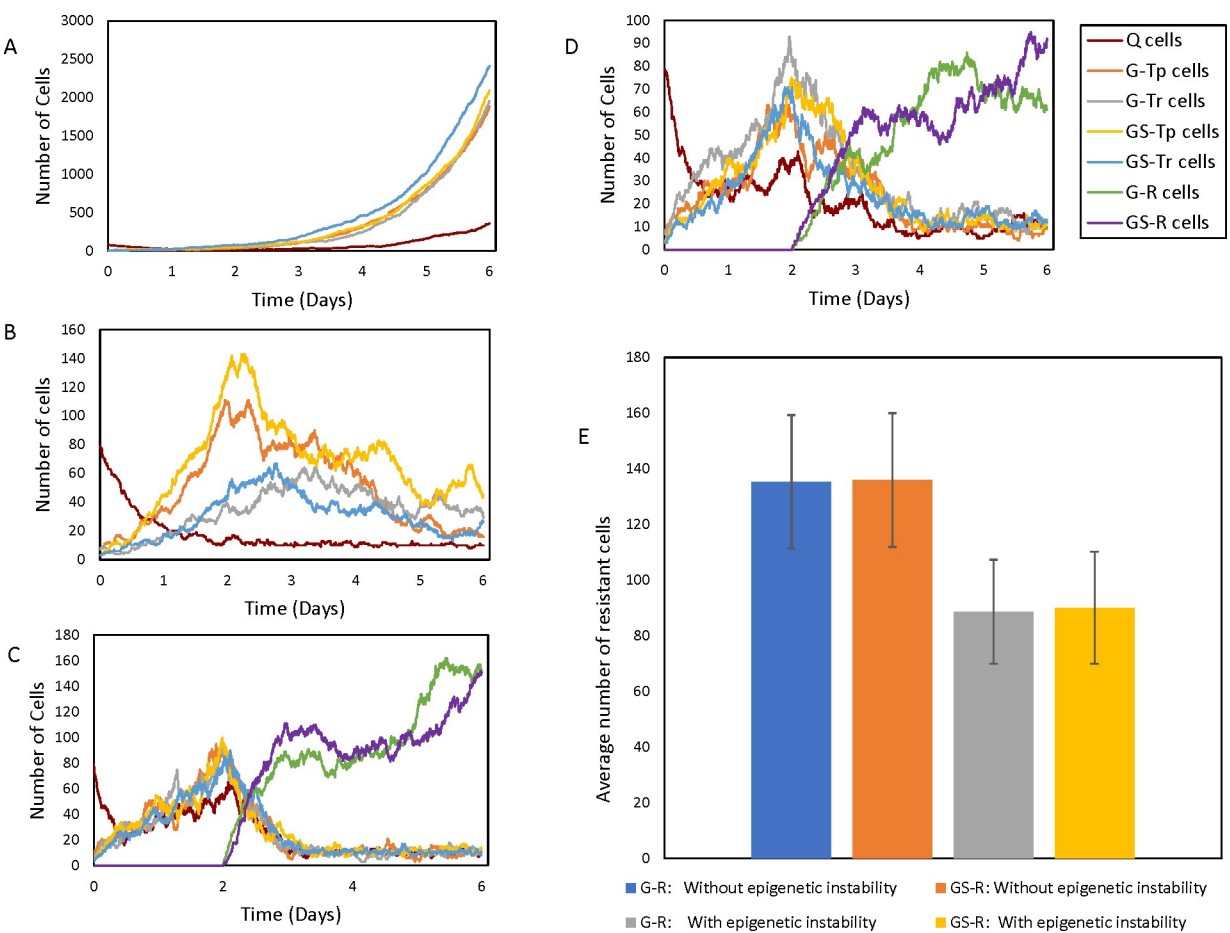

**Fig 2. Cell population dynamics for the basic single-step model.** (A) Control with low transition rates and no drug exposure. (B) Drug-sensitive cells with drug introduced at t = 2 days in the absence of drug resistance. (C) Drug-resistant cells, with sensitive to resistant state transitions enabled upon addition of drug at t = 2 days. (D) Conditions same as C with epigenetic instability allowed upon drug exposure at t = 2 days. (E) Average (± standard deviations) resistant cell numbers at t = 6 days based on 2000 independent simulations. Table 1 lists parameters for each condition.

affect cell state dynamics. The first set of simulations are shown in Fig 2 using the parameters listed in Table 1 that indicate birth, death and cell transition rates for each cell type under different conditions.

**Table 1. Summary of the model simulation parameters.** The various conditions and parameter values for each cell type, where n indicates the number of cells in a state; u is a random number parameter generating instability in the transitions; t is the time elapsed in the system; and $t_{max}$ is the maximum time allowed in the simulations. BR: birth rate, DR: death rate and TR: transition rate.

| Condition | Time (t) | Q Cells | | | G-Tp, G-Tr, GS-Tp, GS-Tr | | | G-R, GS-R | | | Figure Number |
|---|---|---|---|---|---|---|---|---|---|---|---|
| | | BR | DR | TR | BR | DR | TR | BR | DR | TR | |
| Control No drug | | 0 | n | 0.1*n | 2*n | n | 0.1*n | NA | NA | NA | Fig 2A |
| Drug sensitive | t < 2 | 0 | n | 0.1*n | 2*n | n | 0.1*n | NA | NA | NA | Fig 2B |
| | t > 2 | 0 | 2.2*n | 0.1*n | 2*n | 2.2*n | 0.1*n | NA | NA | NA | |
| Drug resistant with stable transitions | t < 2 | 0 | n | n | 2*n | n | n | NA | NA | NA | Fig 2C |
| | t > 2 | 0 | 2.2*n | n | 2*n | 2.2*n | n | 2*n | $2.2*n*e^{\frac{-(t-2)}{t_{max}}}$ | n | |
| Drug resistant with unstable transitions | t < 2 | 0 | n | n*u | 2*n | n | n*u | NA | NA | NA | Fig 2D |
| | t > 2 | 0 | 2.2*n | n*u | 2*n | 2.2*n | n*u | 2*n | $2.2*n*e^{\frac{-(t-2)}{t_{max}}}$ | n*u | |

Under control conditions (without drug) and minimal cell state transitions (Fig 2A), cells grow exponentially without large changes in the population of Q cells. This leads to a dynamically-balanced population where none of the cell states are dominant. In Fig 2B, drug-sensitive cell population dynamics are shown upon the addition of a drug with the condition that drug resistant populations are not allowed. Initially, there is exponential growth similar to the control conditions, followed by a decrease in all cell populations after the drug is introduced at 2 days since there are no transitions into resistant states. In control and drug sensitive cases (Fig 2A and 2B), the transition rates are minimal at all times ($0.1{}^{*}n$). When transitions to drug resistant states are allowed the cell dynamics change accordingly (Fig 2C). Prior to addition of the drug, cells in all states grow exponentially like in Fig 2A and 2B, without any dominant cell state. Now, as drug resistance evolves, all sensitive cell state populations decrease at the expense of an increase in resistant cell state populations. This happens because sensitive cells can transition into resistant cells but the reverse cannot happen. In addition, the death rate of resistant cells decreases with time (due to acquiring resistance) but the birth rate remains the same. Hence, unless the entire population of resistant cells die, they grow exponentially later. Due to the important role of birth and death rates of resistant cells (apart from transition rates among all cell states) in the model dynamics, we investigated their effect on population growth (Table A in S1 Text). As expected, a higher birth rate of resistant cells leads to an increase in their population. Similarly, a higher death rate leads to a decrease in the resistant cell population. A high rate of transition keeps the cell population evenly distributed among the cell states (S1 Fig).

When epigenetic instability is introduced in the transition rates via a stochastic parameter (see Table 1) simultaneously with drug exposure there is a reduction in the overall population of resistant cells due to the random changes in the transition rates (Fig 2D). Since the simulations shown in Fig 2A–2D represent single population trajectories, we performed a large number of simulations (2000) in order to understand the overall behavior of the model (Fig 2E). The average resistant cell population with or without epigenetic instability agree with the results from Fig 2A–2D, and indicate that epigenetic instability reduces the resistant cell population size by 35%.

## Epigenetic instability in the multi-step model

The single-step model may not capture the complex transcriptional machinery due to PTMs and associated changes in chromatin structure, and the requirement of coordinated protein complexes to initiate gene transcription. Thus, a multi-step model was developed that considered the transition from drug-sensitive cells to drug-resistant as a multi-step lock-in process (Fig 3A and 3B). The lock-in process was implemented as a 3-step switch wherein all three events must occur to complete the transition from a sensitive to a resistant state (all switches must be on). The probability of triggering the switch is equal for all switches ($p_1 = p_2 = p_3$, Fig 3B), and further, the triggering probability of each individual switch is independent of the other switches. To apply epigenetic instability, the unidirectional lock-in multi-step process is replaced by a bidirectional multi-step process (Fig 3B), where any of the three switches in an on state, can be turned off.

The control and drug-sensitive conditions for the multi-step model are analogous to the basic model, and thus, only the drug-resistant condition with and without epigenetic fluctuations were considered. The forward multi-step lock-in process leads to drug resistant states from both glioma stem cells and glioma cells (Fig 4A, Top panel). The parameters used to generate the simulations in Fig 4 are the same as those used in Fig 2C; however, in the multi-step model the number of resistant cells is reduced given the more complex lock-in process

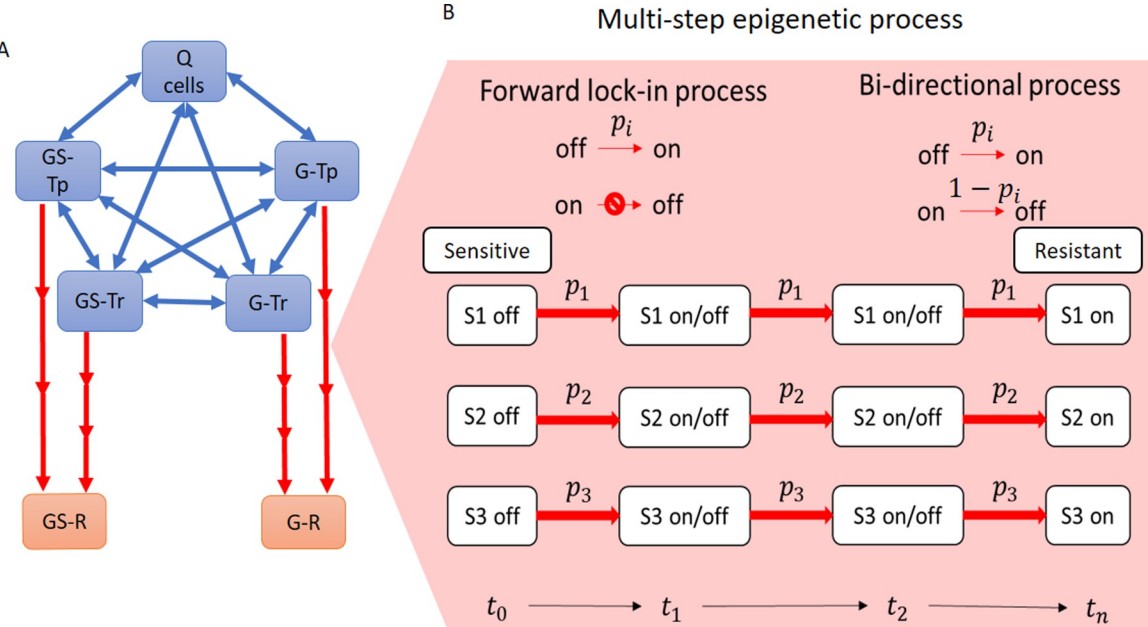

**Fig 3. Schematic of the multi-Step model.** (A) The multi-step model requires transitions to drug-resistant states to be a multi-step process (red arrows). Single-step processes are shown as blue-arrows. (B) The three-step transition model is shown between sensitive and resistant cell states. The forward lock-in process requires all three switches (S1, S2 and S3) to be on, whereas the bidirectional process allows any of the three switches to go from on➜off, thus, interrupting a transition to a resistant state. $p_i$ indicates the probability of the event occurrence (switch turning off➜on or on➜off).

involved in completing a state transition. When epigenetic instability is introduced via the bi-directional process (50% probability of switch turning off➜on and vice versa), the growth of resistant cells is hampered (40% reduction) (Fig 4A, Middle panel). Moreover, high epigenetic instability (80% probability of switch turning on to off), led to a much larger reduction (90%) in resistant cell populations (Fig 4A, Bottom Panel). As with the basic model, conducing a large number of simulations (Fig 4B) indicates that moderate epigenetic instability reduces the resistant cell population by about 60% (see Fig 4B). At high epigenetic disruption, the resistant cell population is reduced by more than 90% compared to the scenario without any epigenetic disruption.

In order to determine the effect of changing levels of epigenetic instability on the resistant cell population, we performed simulations changing the probability of the switches turning on➜off. As shown in Fig 4C, there is an inverse relationship between the probability of a unit switch turning off (a measure of epigenetic instability) and the size of the resistant cell population. At the highest tested probability of epigenetic disruption of 80%, the resistance cell numbers declined by more than 90%.

### Hybrid stochastic-deterministic TMZ-MGMT Model

A pharmacokinetic model of TMZ that accounted for the production of DNA adducts and specifically, O6mG, previously applied to GBM patients [23] was linked to a two-stage gene transcription-translation model for MGMT. This ODE model (see Figs 5 and S3) can simulate the dynamics of each designated species such as MGMT and O6mG following a standard multiple-dose (daily X 5 days) regimen of TMZ. The stochastic cell state transition model (Fig 5) analogous to the multi-step stochastic cell state model was linked to the ODE TMZ-MGMT

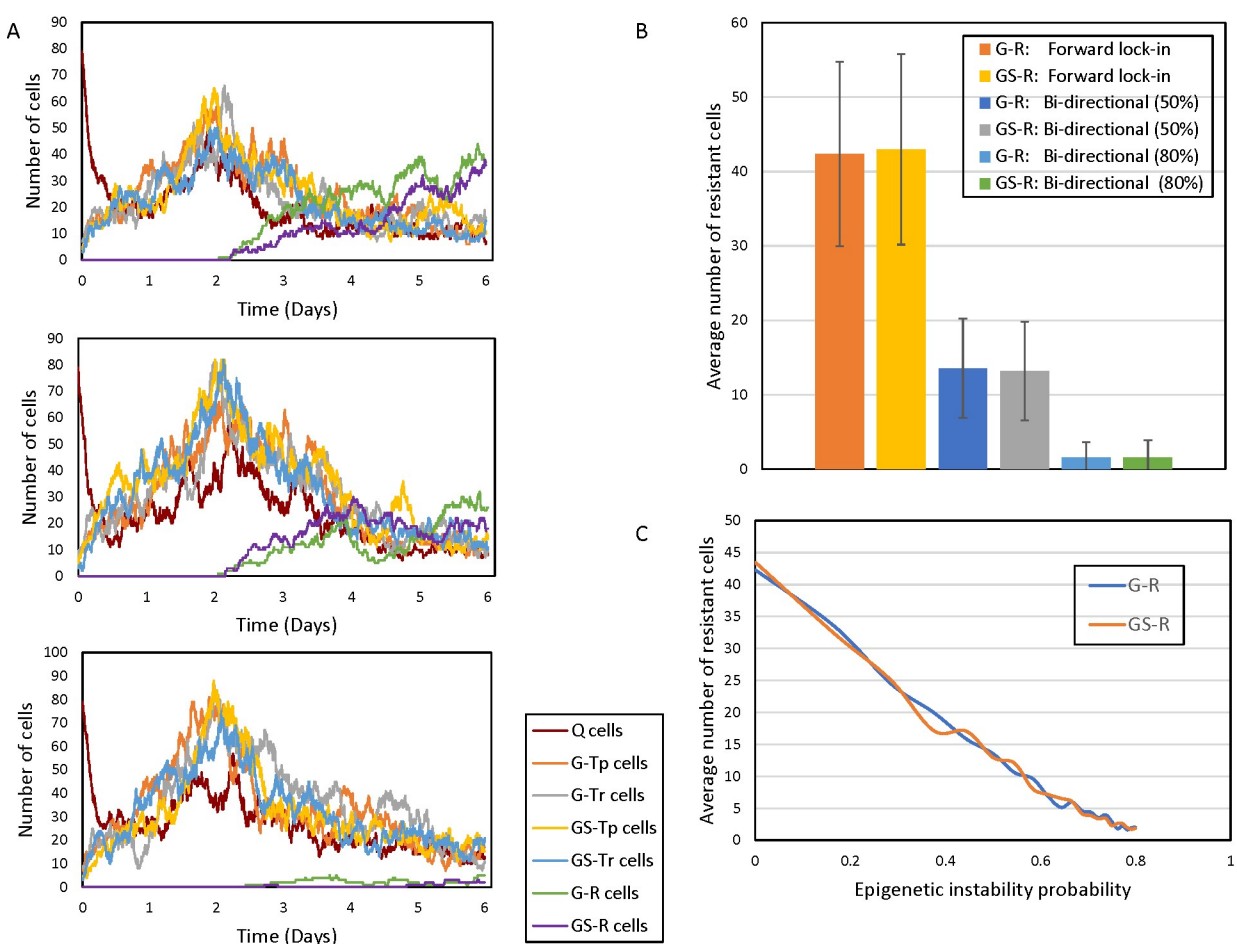

**Fig 4. Cell population dynamics for the multi-step model.** (A) Top panel: Forward lock-in multi-step process with drug introduced at t = 2 days. Middle panel: Bi-directional multi-step process with moderate epigenetic instability (50% probability of switch turning on➜off). Bottom panel: Bi-directional multi-step process with high epigenetic instability (80% probability of switch turning on➜off) (B) Average (± standard deviations) number of resistant cells at t = 6 days following 2000 independent simulations. The values 50% and 80% in the legend are the probability of switch turning on➜off. (C) The average number of resistant cells (GS-R plus G-R) as a function of increasing epigenetic instability (probability of switch turning on➜off) in a multi-step bi-directional process.

model. A key difference is that the cell state model does not explicitly track resistant cell populations but rather specifies G-Tp MGMT and GS-Tp MGMT populations wherein the multi-step epigenetic lock-in process emanates from either the whole population of transcriptionally-permissive cells, both G and GS. It is the transcriptionally-permissive MGMT cells that will determine the final quantities of MGMT and O6mG. MGMT acts as a suicide inhibitor of O6mG [24], and thus, the complex is degraded, and new MGMT is activated via a positive feedback loop to MGMT mRNA (see Fig 5).

It can be seen (Fig 6) that the presence (bi-directional multi-step) of and extent (50% and 80%) of epigenetic instability influences the final populations of G-Tp MGMT and Gs-Tp MGMT cells similar to the resistant cell populations in the totally stochastic cell state models. The simulated MGMT and O6mG concentrations (see Fig 6) under these same conditions (no epigenetic instability or with epigenetic instability at either 50% or 80% probability) yield the anticipated results with high MGMT/low O6mG concentrations in the absence of epigenetic instability and graded low MGMT/high O6mG concentrations depending on whether 50% or

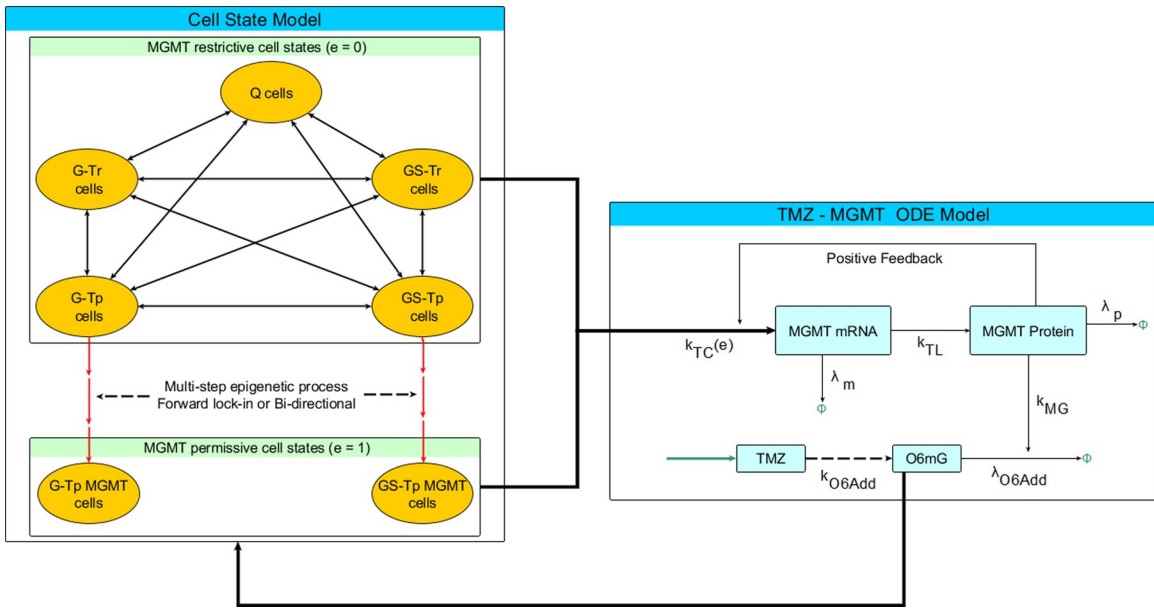

**Fig 5. Schematic of the Hybrid Stochastic-Deterministic TMZ-MGMT model.** The stochastic cell state model–based on the multi-step model (Fig 3)–is modified to distinguish G-Tp MGMT and GS-Tp MGMT cell populations. The multi-step epigenetic process (either forward lock-in or bi-directional, dotted black line) is specifically used for Tp transitions to Tp MGMT transitions, both G-Tp MGMT and GS-Tp MGMT populations. Both the Tr and Tp cell populations are linked to the MGMT ODE model wherein an epigenetic designation of 0 (restrictive) and 1 (permissive) determines the dynamics of MGMT and O6mG. The TMZ dosing schedule is designated by the green input arrow. TMZ undergoes pH-dependent multi-step metabolic conversion to O6mG adducts (indicated by dotted arrow, see S3 Fig). The quantity of O6mG adducts determines the cell death rate that feeds back to the stochastic cell state model. The rate constants associated with the ODE model are defined in Table B in S1 Text and designated in equations S5-S7.

80% epigenetic instability is used. The oscillating O6mG profiles (Fig 6) are due to the multiple-dose TMZ schedule of five doses every 24 hours starting at t = 24 hour.

## Discussion

Anticancer drug resistance has been studied for decades with the main solution being the addition of drugs to circumvent resistance and restore cancer cell drug sensitivity [25–30]. Genetic analyses of gene mutations and expression profiles are often the basis of the drug resistance strategies that seek to identify the drug resistant-inducing genes and proteins, which may provide viable drug targets. Although the approach has appeal given the mechanistic justification, and has produced a plethora of drug combinations, these approaches ultimately fail since they do not consider intratumoral heterogeneity and innate differences in cell sensitivity to drugs, as well as cellular adaptation including the role of epigenetic reprogramming [31].

Both deterministic and stochastic mathematical models have been employed to study intratumoral heterogeneity and the emergence of drug resistance in cancer [32–37]. Among deterministic approaches, the most common use ODEs to model population growth of tumors [38]. Stochastic approaches, although computationally taxing, are generally more appropriate for modeling biological systems, which are naturally noisy and replicate the dynamics of gene transcription. In particular, discrete stochastic processes are quite relevant in modeling tumor growth as they can predict the probability of development of at least one resistant cell in a tumor [39–41]. Stochastic differential equations are used to model tumor heterogeneity as they provide a robust methodology for mechanistic modeling and the control of noise [42,43]. The stochastic modeling approach we employed is common to that used in evolutionary

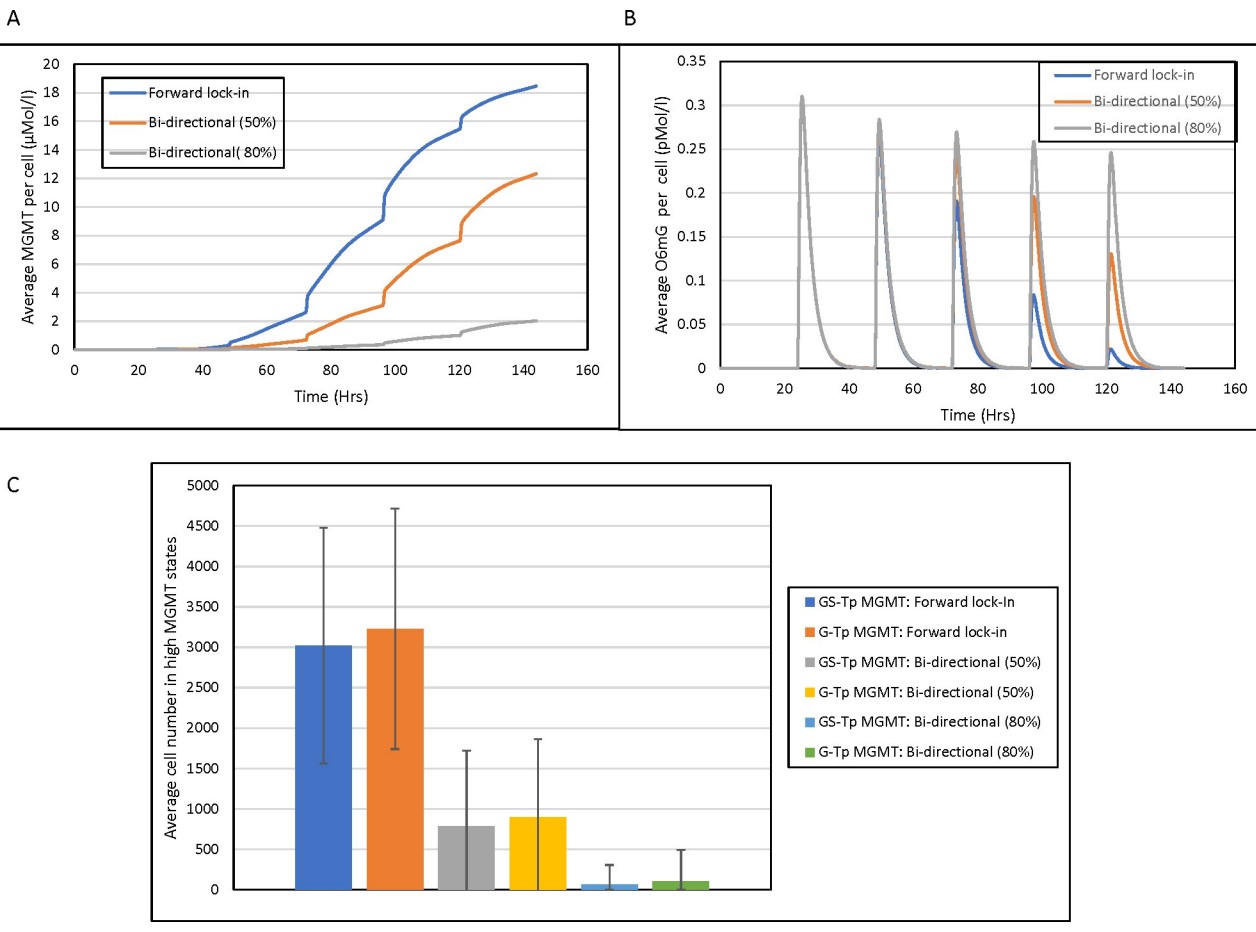

**Fig 6. Cell population dynamics for the hybrid stochastic-deterministic TMZ-MGMT model.** (A) MGMT concentrations following a multiple-dose TMZ schedule (daily X 5 days) starting at 24 hours without epigenetic instability (blue, forward lock-in) and with either 50% (orange, bi-directional 50%) or 80% (gray, bi-directional 80%) epigenetic instability, (B) O6mG concentrations following a multiple-dose TMZ schedule (daily X 5 days) starting at 24 hours without epigenetic instability (blue, forward lock-in) and with either 50% (orange, bi-directional 50%) or 80% (gray, bi-directional 80%) epigenetic instability. (C) GS-Tp MGMT and G-Tp MGMT cell populations (Average cell number at time = 144 hr) under different conditions. The average concentrations (panel A and B) and cell numbers (+SD) are determined from 2000 independent simulations.

dynamics based on branching processes with Markovian properties that are simulated with the Gillespie algorithm [44,45].

In this work, stochastic cell state models were developed to understand the effect of epigenetic fluctuations on tumor heterogeneity and drug resistance. Both the single-step and multi-step models show exponential growth of resistant cells upon drug exposure (Figs 2C and 4A, Top Panel). However, when epigenetic instability was introduced in either model, the resistant cell populations decreased, and were inversely proportional to the probability of the multi-step switches turning off (Fig 4C). In the single-step model, the reduction due to epigenetic instability is about 35%. In the multi-step model, at moderate epigenetic instability (50% probability of a switch turning off), the reduction in resistant cells was about 60% whereas at high epigenetic instability (80% probability of a switch turning off) the reduction is more than 90%.

Both the single-step and multi-step models specify two likely epigenetic states, either transcriptionally permissive or restrictive. In this manner, there is no assigned preference to either state in terms of contributing to or mitigating drug resistant states as the permissive state could produce oncogenes or drug inhibiting proteins, whereas the restrictive state could

prevent activation of tumor suppressors. As such, either transcriptional state has an equal probability to transition to a resistant state.

The incorporation of the mechanistic TMZ-MGMT model with the cell state model allowed us to focus on one transcriptional state, and for MGMT—the DNA repair enzyme—the permissive state would lead to higher MGMT concentrations and elicit its drug-resistant attributes by removing TMZ-generated O6mG DNA adducts. Introduction of epigenetic instability in the transitions to G-Tp MGMT and GS-Tp MGMT cells clearly limited the production of MGMT that lead to higher O6mG adducts.

Our computational models do not address how specifically to induce epigenetic instability. We posit that epigenetic modifiers are the pharmacological tool to achieve epigenetic instability. In this regard, knowledge of histone PTM profiles for specific drug-resistant genes could be a starting point to target for disruption; however, it should be appreciated that the sought-after epigenetic instability has a chaotic component. Thus, epigenetic modifiers that yield the desired action—producing either a transcriptionally-permissive or -restrictive state—should not fall into the trap of constant dosing paradigms, but rather be used in a more random fashion to prevent cellular adaptation. Thus, although epigenetic modifiers are a route to alter histones PTM, their dosing schedules should be variable to support instability and minimize cell adaptation.

Our approach of combating anticancer drug resistance through generation of epigenetic instability may have advantages over prevailing methods. First, it reduces the complexity of ITH at least in terms of epigenetic plasticity as consisting of either transcriptionally-permissive or -restrictive cell states. This simplification treated as an agnostic feature in the stochastic cell state models was tuned to repress MGMT production in the hybrid cell state–TMZ MGMT ODE model. Second, a chaotic or constantly reprogramed epigenome may limit cellular adaptations including those induced by constant drug exposures. Current approaches to aggressively target one or more resistance pathways may select for alternate drug-resistant pathways to emerge. Although the resistant cells are not extinguished at 50% probability of the on/off switch in the multi-step model, they are not proliferating or are doing so very slowly. This attribute of epigenetic instability—maintenance of this quasi-equilibrium between sensitive and drug-resistant cells—could stall or prevent implementation of alternate drug-resistant programs possibly rendering the cells susceptible to other treatments.

The proposed cell state models are theoretical and remain to be experimentally tested. There have been investigations on TMZ-induced resistance in GBM driven by MGMT that delineated a unique histone-mediated mechanism, and further that the HDAC inhibitor SAHA increased H3K9 acetylation (H3K27ac) and enriched MGMT production [46]. There is an effort to develop bromodomain and extraterminal (BET) inhibitors, and specifically, bromodomain protein 4 (BRD4) inhibitors for GBM that can decrease H3K27ac [47]. At least one BRD4 inhibitor, OTX015, has been used in clinical trials including for GBM patients [47]. Whether reversal of H3K9ac or other combinations of histone PTM could prevent MGMT expression remains to be determined.

In conclusion, stochastic and hybrid stochastic-deterministic models were developed that demonstrated epigenetic instability has the potential to mitigate the emergence of drug resistant cells. The integrated cell state-mechanistic TMZ MGMT model serves as a foundation to explore detailed approaches to implement epigenetic modifier therapy. It is important to assess whether global and targeted epigenetic instability through combinations of epigenetic modifiers and non-constant dosing regimens differ in their ability to generate epigenetic instability. These aspirational goals can be realized by use of experimental methods and the computational models as presented here.

## Methods

### General description of cell state model

The mathematical details of the cell state models and how the simulations were conducted are provided in the S1 Text.

In the basic model, only single-step state transitions are allowed (see Fig 1). All state transitions are reversible except those that culminate into a drug resistant state. Once a cell is deemed drug resistant, it cannot transition to any other state. Integer cell numbers are used for each cell state to discretize cell numbers within the tumor. The system dynamics can be written in terms of a master ordinary differential equation that governs the time evolution of the probability of the system occupying a given cell state (S2 Fig).

We incorporate stochasticity in our model by using a Monte Carlo methodology to simulate birth, death, and state transitions. As quiescent cells do not proliferate, the birth rate of Q cells is set to be zero. The death rates of quiescent cells were chosen to be a proportional to the number of cells in the state and to incorporate the effect of drug, the death rate coefficients increases after the addition of drug (See Table 1). Additionally, the Q cell population can increase or decrease due to state transitions and the transition rates were set to be proportional to the number of cells in the state from which the cell is transitioning.

For G and GS cell states, birth/death and transition rates are proportional to the number of cells in that state. Before the addition of drug, the birth rate coefficient is set higher than the death rate coefficient, which leads to exponential growth. After the addition of drugs, the death rate is set higher than the birth rate to account for the effects of drug, which leads to the decline in sensitive cells (Q cells, G and GS) populations. Moreover, the birth rate and initial death rate of the drug resistant states are kept the same as drug sensitive states. However, due to acquired drug resistance, the death rate of resistant cells decays exponentially with time.

We considered cell transitions to resistant states under two different models. In the basic model (see Fig 1), single-step one-way transitions occur from a drug sensitive state to a resistant state. In the multi-step model (see Fig 3), three uni-directional steps are required to "lock-in" a transition from a sensitive to a resistant cell state. In both the basic and multi-step models, two cases—with and without epigenetic instability–are considered.

For the simulations, we keep the initial total number of cells = 100, among which Q = 80 GS-Tp = 3 GS-Tr = 3 GS-R = 0 G-Tp = 7 G-Tr = 7 G-R = 0. The drug is introduced in the system at time t = 2 days. In the single step model, epigenetic instability or fluctuations in the transition rate is modeled using a random number. All relevant parameters regarding each simulation is given in Table 1. To avoid complete elimination of the sensitive cell population, we apply constraints on the death rate and transition rates. If any sensitive cell state population is less than 10, then the death rate and transition rate to resistant cells is reduced to zero. All rate coefficients have units Day$^{-1}$. A maximum time ($t_{max}$) of 20 days is allowed during simulations. Additional details regarding the implementation of Gillespie algorithm are provided in the S1 Text.

### Hybrid stochastic-deterministic TMZ-MGMT Model

The model is an amalgam of a deterministic ODE TMZ—MGMT model which determines the MGMT mRNA, MGMT protein and O6mG levels in the cell, and a stochastic cell state transition model which describes the dynamics at the cell population level. The stochastic cell state component of the model is similar to the multi-step cell state model developed earlier (Fig 3) with the exception that only G-Tp/GS-Tp can transition to the corresponding G-Tp/

GS-Tp MGMT cell states (Fig 5). Hence, we considered two cases: one with and the other without epigenetic instability. Birth and transition rates are taken from the multi-step cell state model. However, the death rates in the cell state model depend on the O6mG adduct concentrations which are generated from the TMZ-MGMT ODE model.

Regarding the deterministic TMZ-MGMT component, the TMZ dosing schedule was 150 mg/mm$^2$ every 24 hr for 5 doses as this is a standard of care regimen in GBM patients [21,48]. The pharmacokinetic model was based on the work of Ballesta et al [23] that accounted for the pH-dependent conversion of TMZ to a sequential series of metabolites that terminated with the production of O6mG (see S3 Fig). Both the relevant parameters regarding the MGMT transcription and translation, and the MGMT-O6mG complex degradation rate constant were taken from the literature [49,50]. The MGMT mRNA and protein values were at steady-state prior to TMZ administration as determined by the MGMT ODE model in the absence of TMZ. The detailed ODE model can be found in the S1 Text.

## Supporting information

**S1 Fig. Effects of changing birth/death/transition rates on cell population dynamics.** (A) Simulation for low birth rate of resistant cells (1.5*n). (B) Simulation for high birth rate of resistant cells (2.5*n). (C) Simulation for low death rate of resistant cells (2.2*n* e$^{-(t-2)/10}$). (D) Simulation for high death rate of resistant cells (2.2*n* e$^{-(t-2)/100}$). (E) Simulation without any transition. (F) Simulation with high transition rate (2*n).
(TIFF)

**S2 Fig. Cell state probability flowchart.** Illustration of the flow of cell state probability within the cell state space through the process of birth, death, and transition. $n_i$ Indicates the number of cells in the $i^{th}$ state. $c_{bi}$, $c_{di}$ and $c_{t_{ij}}$ are birth, death and transition rate coefficient, respectively.
(TIF)

**S3 Fig. TMZ PK Model.** (A) The TMZ PK model based on that developed by Ballesta et al [1] that shows the intracellular metabolic conversion from TMZ to O6mG. An oral 150 mg/m$^2$ TMZ dose every 24 hours for 5 days starting at t = 24 hours produced concentration-time profiles of TMZ in blood (B), interstitial fluid TMZ (C), intracellular TMZ (D), methylating cation (E), and O6mG (F). The O6mG concentrations are greater than in Fig 6 since there is no MGMT repair considered here.
(TIFF)

**S1 Text. Supplement.**
(DOCX)

## Author Contributions

**Conceptualization:** Anshul Saini, James M. Gallo.

**Formal analysis:** Anshul Saini, James M. Gallo.

**Methodology:** Anshul Saini, James M. Gallo.

**Software:** Anshul Saini.

**Writing – original draft:** Anshul Saini, James M. Gallo.

**Writing – review & editing:** Anshul Saini, James M. Gallo.

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
