## [Decision Letter · Decision Letter 0]

8 Apr 2021

Dear Dr. Gallo,

Thank you very much for submitting your manuscript "Epigenetic instability may alter cell state transitions and anticancer drug resistance" for consideration at PLOS Computational Biology.

As with all papers reviewed by the journal, your manuscript was reviewed by members of the editorial board and by several independent reviewers. In light of the reviews (below this email), we would like to invite the resubmission of a significantly-revised version that takes into account the reviewers' comments.

We cannot make any decision about publication until we have seen the revised manuscript and your response to the reviewers' comments. Your revised manuscript is also likely to be sent to reviewers for further evaluation.

Sincerely,

Ilya Ioshikhes

Deputy Editor

PLOS Computational Biology

Douglas Lauffenburger

Deputy Editor

PLOS Computational Biology

Reviewer's Responses to Questions

**Comments to the Authors:**

Reviewer #1: The authors have developed stochastic models for cancer cell population that capture transition among various states that are capable of switching among multiple cell states, and in the presence of drug, can undergo an irreversible transition to a drug-resistant state. They investigate the impact of introducing epigenetic instability in their model and quantify the effect on population state distribution under many conditions and suggest mechanisms to potentially reduce the dominance of drug-resistant population. The study endeavours to ask important questions for therapeutic management of cancer; however, it seems to falls short on both fronts in its current form – a) comparing their model to experimental data that may be available for combinatorial therapies (not necessarily in gliomas though) and b) generating specific predictions that can be experimentally tested. Thus, the authors need to conduct additional analysis before the manuscript can be considered for publication:

1. The authors should discuss more of the relevant literature in terms of modeling the effects of epigenetic on cell-fate decisions (Miyamoto et al. PLoS Comp Biol 2015; Jia et al. Phys Biol 2019) and its effect in designing epigenetic-targeting drugs (Gunnarsson et al. J Theor Biol 2020). They should discuss conceptual and technical similarities and differences in their work vis-à-vis previous literature.

2. Cell-state transitions are also possible during cell division (Tripathi et al. PLoS Comp Biol 2020). How does incorporating such effects alter the behavior of the models shown here?

3. Have the authors investigated the impact of having intermittent therapy (i.e. drug holidays) which has been included as a part of various clinical trials now? (Michor & Beal, Cell 2015)

4. What is so specific about this model in terms of gliomas, as the authors claim?

5. Is the only major difference between the first model and the multi-step model that there is one more intermediate step that needs to be completed before the lock-in? If true, how do the results of the model change when more than one such intermediaries are included? Also, how do the effects of bidirectional transitions aggregate in terms of such n-state multi-state models? Can bidirectional transitions and multi-state modes sufficient to largely delay tumor resurgence?

Reviewer #2: Modeling without experimental validation is just modeling.

Reviewer #3: The paper needs to be re-written for certain sections as it is not very clear to the reader at the first instance. Also the proposed model needs to be reconsidered as to how much does this statistical model help in understanding drug resistance with respect to other aspects apart from epigenetic instability. Moreover, citing other published articles on the same lines in the manuscript will be appreciated. The idea is good but the authors need to answer certain questions: 1) Is the concept novel or are their other papers that have been published on the same lines been published earlier? 2) How will epigenetic instability in terms of dysregulation alter the drug resistance mechanism? 3) Have the author thought of doing certain experiments to justify their results in terms of any in vitro drug resistance model and epigenetic instability.

Reviewer #4: Authors focus on the most important problem that affects anticancer therapy which is resistance. In this manuscript, the authors offer a new approach focused on causing epigenetic instability to cancer cells. Authors developed a mathematical model of glioblastoma cells that can transition from one state to another based on cell stresses, such as drug therapy and epigenetic plasticity. According to this model, when epigenetically instability was allowed, the number of resistant cells decreased.

Introduction should be improved:

It is important to give information about the epigenetic modifications that authors will introduce in their model, arguing why they decided to introduce histone post-translational modifications instead of other types of epigenetic modifications (DNA methylation, DNA acetylation, etc.).

Authors should also give information about the stochastic cell state models considering permissive or restrictive states.

Discussion should be improved:

It is also important to discuss up to what point the introduction of histone post-translational modifications will not affect health tissue, or which could be the effect of increasing epigenetic instability in health tissue. It would be interesting that authors present a model where epigenetic instability is introduced and increased in normal cells and to report what happens in these cells.

**Have all data underlying the figures and results presented in the manuscript been provided?**

Reviewer #1: Yes

Reviewer #2: Yes

Reviewer #3: Yes

PLOS authors have the option to publish the peer review history of their article (what does this mean?). If published, this will include your full peer review and any attached files.

Reviewer #1: No

Reviewer #2: No

Reviewer #3: No

Reviewer #4: No

**Have the authors made all data and (if applicable) computational code underlying the findings in their manuscript fully available?**

Reviewer #4: **No: **Authors did not introduce in the manuscript the statement of data availability.
---

## [Decision Letter · Decision Letter 1]

26 Jul 2021

Dear Dr. Gallo,

We are pleased to inform you that your manuscript 'Epigenetic instability may alter cell state transitions and anticancer drug resistance' has been provisionally accepted for publication in PLOS Computational Biology.

Best regards,

Ilya Ioshikhes

Deputy Editor

PLOS Computational Biology

Douglas Lauffenburger

Deputy Editor

PLOS Computational Biology

Reviewer's Responses to Questions

**Comments to the Authors:**

Reviewer #1: The authors have addressed my comments.

Reviewer #3: The revisions have been properly addressed and the necessary changes have been made accordingly. Moreover, one suggestion that I would like to give the authors is to back up their model with relevant experimental work that can be published later.

**Have the authors made all data and (if applicable) computational code underlying the findings in their manuscript fully available?**

Reviewer #1: Yes

Reviewer #3: Yes

PLOS authors have the option to publish the peer review history of their article (what does this mean?). If published, this will include your full peer review and any attached files.

Reviewer #1: No

Reviewer #3: No

---

## [Editor Report · Acceptance letter]

16 Aug 2021

PCOMPBIOL-D-20-02125R1 

Epigenetic instability may alter cell state transitions and anticancer drug resistance

Dear Dr Gallo,

I am pleased to inform you that your manuscript has been formally accepted for publication in PLOS Computational Biology. Your manuscript is now with our production department and you will be notified of the publication date in due course.

With kind regards,

Olena Szabo
